# The MosHouse^®^ Trap: Evaluation of the Efficiency in Trapping Sterile *Aedes aegypti* Males in Semi-Field Conditions

**DOI:** 10.3390/insects13111050

**Published:** 2022-11-14

**Authors:** Pattamaporn Kittayapong, Rungrith Kittayapong, Suwannapa Ninphanomchai, Wanitch Limohpasmanee

**Affiliations:** 1Center of Excellence for Vectors and Vector-Borne Diseases, Faculty of Science, Mahidol University at Salaya, Nakhon Pathom 73170, Thailand; 2Department of Biology, Faculty of Science, Mahidol University, Bangkok 10400, Thailand; 3Go Green Company Limited, Science Building 2, Faculty of Science, Mahidol University at Salaya, Nakhon Pathom 73170, Thailand; 4Thailand Institute of Nuclear Technology, Ministry of Higher Education, Science, Research and Innovation, Nakhon Nayok 26120, Thailand

**Keywords:** *Aedes aegypti*, mosquito trap, mosquito control, vector surveillance tool, sterile insect technique, incompatible insect technique

## Abstract

**Simple Summary:**

Trapping mosquitoes, especially those that are vectors, is important in evaluating disease control programs. So far, mosquito-collecting tools that are inexpensive and highly effective in collecting *Aedes aegypti*, the main mosquito vector of dengue, chikungunya, and Zika viruses, are not available. In particular, male trapping is necessary for monitoring control efforts that use released sterile males. In this study, we evaluate the efficiency of a simple and low-cost MosHouse sticky trap in collecting *Ae. aegypti* in a semi-field condition. When comparing the MosHouse traps with the Biogents’ BG-Sentinel traps, which are widely used for collecting *Aedes* mosquitoes, the results showed no significant difference in the numbers of collected males but significantly lower numbers of females were collected using the MosHouse traps. We also found that sterilizing males by radiation significantly increased their collection when using the MosHouse traps. Improvements were made to the MosHouse trap to increase male collection by adding a sugar stick and sticky flags—the latter increasing the number of trapped males but not females when they were released separately, while the number of both males and females increased when they were released together. In summary, the MosHouse trap was proved to be efficient and could be used as an alternative collecting tool in *Ae. aegypti* control programs.

**Abstract:**

Arbovirus diseases, such as dengue, chikungunya, and Zika, are important public health problems. Controlling the major vector, *Aedes aegypti*, is the only approach to suppressing these diseases. The surveillance of this mosquito species needs effective collecting methods. In this study, a simple MosHouse sticky trap was evaluated in a semi-field condition. Our results demonstrated the efficiency of this trap in collecting *Ae. aegypti* males, and no significant difference (*p* > 0.05) in the numbers of males was detected when compared with the widely used BG- Sentinel trap. However, there were significantly lower numbers of females (*p* < 0.05) collected using the MosHouse trap when compared to the BG-Sentinel trap. We also found a significant difference (*p* < 0.05) in the collected numbers between irradiated and non-irradiated males. More irradiated males were collected in the MosHouse traps. The improvement of male collection was achieved with the addition of a sugar stick and sticky flags. Significantly higher numbers of males were collected in the MosHouse trap with sticky flags compared to the original one when they were released independently of females, but both were collected in higher numbers when they were released together (*p* < 0.05). In conclusion, our experiments demonstrated that the MosHouse trap could sample *Ae. aegypti*, especially males, as efficiently as the established BG-Sentinel trap, while the cost was more than 50 times lower, showing the potential of the MosHouse trap for improved *Ae. aegypti* male and female surveillance with very large numbers of traps at affordable costs. In addition, significantly (*p* < 0.001) increased male sampling was achieved by adding an external sticky flag on the MosHouse trap, providing an avenue for further development of the novel male-trapping strategy.

## 1. Introduction

Arbovirus diseases, such as dengue, chikungunya, and Zika, are a major public health problem in many tropical and subtropical countries. Mosquito vector control is a key to interrupting the transmission of these diseases [1] and collecting sufficient numbers of adult mosquito vectors is necessary in order to understand disease transmission dynamics and provide an appropriate control strategy [2]. In addition, well-designed surveillance programs for both mosquito vectors and pathogens are required to effectively manage these diseases [3].

Several mosquito traps have been developed worldwide in order to sample adult mosquitoes [1] and to increase the control efforts against major mosquito-borne diseases such as malaria, dengue, filariasis, Zika, and Yellow fever [4]. Different trapping methods vary in efficiency and the labor required [1,5,6]. Monitoring the abundance of adult mosquito vectors by evaluating their distribution and density are important aspects of the development of control strategies [7,8]. Since the mosquito traps and trapping techniques that are commercially available have mostly been deployed to reduce nuisance populations of adult mosquitoes [5,9], the efficient trapping of mosquito vectors is still urgently needed [2,10].

Currently, various methods have been applied for the monitoring and control of *Aedes aegypti,* but so far there have been no methods that are able to precisely estimate the density of *Ae. aegypti* adult populations [7]. In general, *Ae. aegypti* mosquitoes, which are the main vector of dengue virus, breed and live inside households or within the surrounding environment [11,12]. In addition, they usually do not disperse very far from where they emerge given that blood meal sources are always available [13]. Based on these facts, the ideal trap to estimate the population density of this domestic mosquito species should be placed inside households or within the surrounding environment. Collection techniques such as the backpack aspirator and BG-Sentinel trap (Biogents, Regensburg, Germany) have shown reliable results [14,15,16], and the latter is reported as one of the most commonly used for the surveillance trapping of adult *Aedes* mosquitoes because it is more sensitive in detecting *Aedes* populations than other available traps on the market [17,18,19,20], especially when the collection is in urban settings [21]. However, since the BG-Sentinel trap requires a power source to supply the electric fan motor that pulls mosquitoes into a collection bag, its deployment depends on large batteries or power sources which are costly and spatially limited [18,19,22,23,24]. Additional concerns when operating BG-Sentinel traps include the need of a CO_2_/lure release and daily activation/maintenance, as well as the trap’s large size and high individual labor cost [6,22]. Although some studies have shown that BG-Sentinel traps were commonly used without CO_2_ for *Ae. aegypti* surveillance [25,26], especially during large-scale field operations in order to reduce additional costs and logistical limitations imposed by CO_2_ incorporation [27], various aspects may influence the manual collection results of these traps, i.e., the competency of the operators, the location of the collection (e.g., indoors vs. outdoors), the size of the collection site, the presence of furniture, and the duration of the sampling [6,27,28]. Therefore, there are some shortcomings of the BG-Sentinel trap when used in mass trapping programs, particularly in developing countries where financial resources are limited. Moreover, the BG-Sentinel trap was not considered cost-effective for daily mosquito collection in endemic areas [6,19,23,29,30,31].

Recently, the use of collection techniques such as ovitraps and sticky traps have been desirable in order to enhance the efficiency of the traps for the optimal monitoring of species abundance, for the assessment of the risk of arbovirus transmission, and for the optimization of vector control activities [18]. Previous studies have highlighted the advantage of using low-cost materials to develop traps for monitoring adult *Ae. aegypti* and *Ae. albopictus* [7]. Due to the low-cost materials used in building sticky traps and the ease in directly identifying and counting the number of adult mosquitoes that enter the traps, this type of trap has been widely used for dengue vector surveillance, the evaluation of the effectiveness of vector control strategies, the monitoring of mosquito population dynamics, and the investigation of the ecological parameters of *Ae. aegypti* in relation to eco-climatic factors [6,28,32,33,34,35]. Various sticky designs have been developed and evaluated in the field condition in order to monitor vector abundance and to analyze the association between adult abundance and risk of dengue transmission [23,28,36,37,38,39,40]. In terms of the designs of sticky traps, there are various models, such as the sticky ovitrap, with an adhesive surface placed on the inner wall of the trap [23,35,40]; the new model of sticky trap developed by Facchinelli et al. [28] that aims to capture adult gravid female mosquitoes; and the three adhesive traps developed by Capuno et al. [23] that have been used to monitor mosquito adult abundance and seasonal dynamics. However, no available trapping methods are specifically designed for male mosquitoes, especially *Ae. aegypti* males.

Trapping methods for male mosquitoes are necessary in the field for the Sterile Insect Technique (SIT) and other control strategies that emphasize the use of male mosquitoes. Since available traps have focused mainly on females, the modification and adjustment of these available trapping methods are needed to fulfill the requirements of cost-effectiveness, ease of trap deployment, less maintenance, etc., for male collection. In this study, the MosHouse trap was used as an available passive trap, which provides resting sites and attracts *Ae. aegypti* mosquitoes by its color, odor, moisture and darkness inside. We conducted a series of experiments using the MosHouse trap, which had been originally designed to attract *Ae. aegypti* females, as well as a modified MosHouse trap that included either a sugar stick or sticky flags, in order to increase the capture rate of *Ae. aegypti* males. The experiments were performed under semi-field conditions in order to evaluate the efficiency of the MosHouse trap and its modifications, and to compare its efficiency in capturing *Ae. aegypti* males and females against the most widely used BG sentinel trap in similar conditions.

## 2. Materials and Methods 

### 2.1. Mosquito Strain

The *Aedes aegypti* strain used in the experiments was originally collected as eggs in communities in Chatuchak, Bangkok, Thailand. The larvae were reared in an insectary maintained at 27 ± 2 °C temperature with 75 ± 2% humidity, and a photoperiod of L12:D12. Pupae were then sex-separated by using a glass separator (John W. Hock, Model 5412, Gainesville, FL, USA). Then they were placed in plastic containers, each 122.66 cm^3^ in volume (diameter 12.5 cm, height 14.5 cm) and with water 62 cm^3^ in volume, prior to transportation to the radiation source. These plastic containers filled with pupae were transported by air-conditioned car from an insectary at Mahidol University, Salaya Campus, Nakhon Pathom Province to the Thailand Institute of Nuclear Technology (Public Organization) (TINT), Nakhon Nayok Province, which is located 112 km. away. Using a Colbalt-60 (Gammar Chamber 5000, Board of Radiation and Isotope Technology (BRIT), DAE, Mumbai, India), an irradiation dose of 50 Gy for 45 s was applied by a qualified staff at TINT. After irradiation, pupae were placed in a plastic container prior to adult emergence, and a 10% sucrose solution mixed with 0.3% Rhodamine B was provided in order to mark the irradiated mosquitoes. After emergence, adults were counted and transferred into the new plastic container using a mouth aspirator with a total of 100 mosquitoes per container. Males and females were separately transferred into different containers prior to release in the screen tents in each experiment. Males and females that emerged from the same batch of pupae that were not irradiated were used for control experiments.

### 2.2. Description of the MosHouse Trap

The MosHouse trap is an adult mosquito trap that receives its name from its external house shape (Figure 1). MosHouse attracts different mosquito species, especially *Ae. aegypti*, by its combined black and red color, the moisture and darkness created inside, and the odor from hay infusion. Once inside the MosHouse trap, mosquitoes have a high chance of becoming attached to the double-sided sticky panel hanging in the middle, under which is a recycled paper tray full of a combination of jelly polymer (to provide humidity after absorbing water), hay infusion (to create odor) and Bti (*Bacillus thuringiensis israelensis*) granules (to kill any emerging larvae). MosHouse is a simple trapping method that does not require an electric power supply or batteries and could be produced at low cost, i.e., as low as US$4 per trap. The trap is made of recycled paper in the shape of a house. When the trap is folded in the designed packaging, the dimensions are 13.5 cm wide × 32.5 cm long × 2.0 cm high. It has a very light weight (153.37 g per package) and could be stacked within limited space for field transportation (Figure 1). MosHouse traps can be used as long as the double-sided sticky panel is not fully stuck with mosquitoes (which usually takes several months). Furthermore, the double-sided sticky panel could be replaced, hence prolonging the lifetime of the trap. The MosHouse was designed and developed by R. Kittayapong and P. Kittayapong, the founders of Go Green Co., Ltd. It won the Second Award of the Design Innovation Contest organized by the National Innovation Agency (NIA) of Thailand in 2009.

### 2.3. Semi-Field Experimental Designs Using the MosHouse Trap

In this study, we further developed and tested the MosHouse trap for trapping *Aedes* male and female mosquitoes in a semi-field setting in Thailand. The original and newly modified MosHouse traps were tested in 5 m × 5 m screened tents under natural conditions in the shade in March 2018 in Chachoengsao Province, eastern Thailand with the temperature of approximately 28–30 °C and the humidity of approximately 70–75%. A series of experiments were conducted to test the following hypotheses: (1) whether there is any difference in the efficiency in collecting *Ae. aegypti* males and females between the MosHouse trap and the BG Sentinel trap; (2) whether there is any difference in the efficiency of the MosHouse trap in collecting either irradiated or non-irradiated *Ae. aegypti* males; (3) whether the sugar stick, which consisted of cotton soaked with 10% sugar solution, could increase the capture rate of irradiated or non-irradiated *Ae. aegypti* males; (4) whether the double-sided sticky flags could increase the capture rate of *Ae. aegypti* males; and (5) whether there is any difference in the number of *Ae. aegypti* males and females collected when both are released together or released separately in a semi-field condition, while providing the MosHouse traps with additional sticky flags.

To improve the MosHouse trap for male mosquito collection, first the sugar stick was added to the trap, in order to lure *Ae. aegypti* male mosquitoes. The sugar stick was placed in a paper cup that was then placed in the middle of the paper tray in the MosHouse after the jelly polymer hardened (Figure 1B). Secondly, two triangular-shaped double-sided sticky flags were added on top and at the corner outside of the MosHouse to enhance the capture of flying *Ae. aegypti* males (Figure 1A). A series of experiments were conducted with an additional sugar stick and sticky flags to determine whether they increased the number of *Ae. aegypti* males entering into these modified MosHouse traps. 

#### 2.3.1. Experiment 1: *Aedes aegypti* Male and Female Sampling with Original MosHouse Trap vs. BG-Sentinel Trap

A total of 100, 3–5-day-old, 24-hour-unfed, non-irradiated *Ae. aegypti* males were released in the screened tents where one BG-Sentinel trap and one original MosHouse trap were placed in the same individual tents at the same time. Mosquitoes were then captured using the BG-Sentinel trap and the original MosHouse trap. The BG-Sentinel trap used in this experiment was only the trap that was equipped with the fan without any lure or baits inside. A direct comparison of the efficiency of the MosHouse trap and the BG-Sentinel trap was conducted with a one-to-one trap ratio. After 24 h, the sticky panels hanging inside each MosHouse trap and the collecting bags inside each BG-Sentinel trap were collected, and the number of mosquitoes was determined. Any male mosquitoes that remained in the screened tents were collected by using portable vacuum aspirators (Go Green Co., Ltd., Chachoengsao, Thailand). Each experiment was conducted in 4 replicates with the rotation of traps in a Latin Square design and the data were statistically analyzed.

#### 2.3.2. Experiment 2: *Aedes aegypti* Male Sampling with MosHouse Trap or MosHouse Trap with Sugar Stick—Comparing Irradiated vs. Non-Irradiated

A total of 100, 3–5-day-old, 24-hour-unfed, irradiated *Ae. aegypti* males were released in the screened tents where one mosquito trap, either the original MosHouse traps or the MosHouse trap with a sugar stick as an attractant, was placed inside per replicate. In parallel, 100, 3–5-day-old, 24-hour-unfed, non-irradiated *Ae. aegypti* males were also released in different tents with the same conditions as the control. Each experiment was conducted in 4 replicates in different screened tents at the same time. For each replicate, the sticky panels inside the MosHouse traps were individually inspected after 24 h. Any male mosquitoes that remained in the screened tents were collected using portable vacuum aspirators.

#### 2.3.3. Experiment 3: *Aedes aegypti* Male Sampling with MosHouse Trap vs. MosHouse Tap with Sugar Stick—Effect of Internal Sugar Stick on Capture of Irradiated or Non-Irradiated Males

The same experiment as previously described in Experiment 2 was conducted and the effect of the sugar stick in collecting either irradiated or non-irradiated males was indirectly compared between the original MosHouse trap and the MosHouse trap with a sugar stick.

#### 2.3.4. Experiment 4: Improved *Aedes aegypti* Male Sampling with Modified MosHouse Trap—Effect of Internal Sugar Stick and External Sticky Flag

Experiments were conducted in a semi-field condition using three types of MosHouse traps: (1) the original MosHouse, which contained a double-sided sticky panel; (2) a MosHouse with an additional sugar stick as an attractant; and (3) a MosHouse with additional double-sided sticky flags. A total number of 100, 24-hour-unfed, non-irradiated male *Ae. aegypti* mosquitoes, aged 3–5 days, were released in the screened tents where one of the three described MosHouse traps was placed in the middle of the tent for each experiment. After 24 h, the sticky panel hanging inside the MosHouse trap and the external sticky flags were collected, and the number of males that entered into the MosHouse trap and became struck on the sticky panel and those that became struck on the external sticky flags was determined. Any male mosquitoes that remained in the screened tents were collected using portable vacuum aspirators. Each experiment was conducted in 4 replicates in different screened tents at the same time.

#### 2.3.5. Experiment 5: *Aedes aegypti* Male and Female Sampling Using MosHouse Trap with External Sticky Flag—Effect of Sexes Released Separately

A total of 100, 3–5-day-old, 24-hour-unfed, non-irradiated *Ae. aegypti* males were released in the screen tents where one MosHouse trap with the external sticky flags was placed in the middle of the tent for each experiment. The same experiment was conducted in parallel where one original MosHouse trap was placed in the middle of the tent and used as a control. Collections of *Ae. aegypti* males, using the MosHouse trap with external sticky flags and the original MosHouse trap, were carried out after 24 h. Any male mosquitoes that remained in the screened tents were collected using portable vacuum aspirators. The same experiment was conducted with 3–5-day-old, 24-hour-unfed, non-irradiated *Ae. aegypti* females, with an evaluation of the number of females on the sticky panel and sticky flags after 24 h. Each experiment was conducted in 4 replicates in different screened tents at the same time.

#### 2.3.6. Experiment 6: *Aedes aegypti* Male and Female Sampling Using MosHouse Trap with External Sticky Flag—Effect of Sexes Released Simultaneously

A total of 50, 3–5-day-old, 24-hour-unfed, non-irradiated *Ae. aegypti* males and females were released together with a total number of 100 mosquitoes in the screen tents where one MosHouse trap with external sticky flags was placed in the middle of the tent. The same experiment was conducted in parallel where the original MosHouse trap was placed inside the tent as a control. After 24 h, the sticky panels hanging inside each MosHouse trap were collected, and the number of males and females that entered into all MosHouse traps and became struck on the sticky panels and external sticky flags was determined. Any mosquitoes that remained in the screened tents were collected using portable vacuum aspirators. Each experiment was conducted in 4 replicates in different screened tents at the same time.

### 2.4. Statistical Analysis

All statistical analyses were performed using SPSS 18.0, Mahidol University License (SPSS Inc., Chicago, IL, USA). The number of mosquitoes collected from each experiment was quasi-normally distributed and analyzed by using the two-way ANOVA. A significance level of 0.05 was used to determine statistical significance.

## 3. Results

### 3.1. Aedes aegypti Male and Female Sampling with MosHouse Trap vs. BG-Sentinel Trap 

In this experiment, the results from the two-way ANOVA showed that the type of mosquito trap (*p* < 0.001) (Table 1), the sex of *Ae. aegypti* mosquitoes (*p* < 0.001), the position of the mosquito trap (*p* < 0.001), and the interaction of these three factors significantly affected the collection numbers of *Ae. aegypti* (*p* < 0.001) (Table 1). However, when focused on *Ae. aegypti* males, the results indicated that the type of mosquito trap had no effect on collecting males (*p* = 0.063). Figure 2 illustrates the lack of significant differences in the numbers of males collected using the BG-Sentinel trap when compared to those collected using the MosHouse trap (44.50 ± 18.92 vs. 40.00 ± 15.26). On the contrary, when focused on females, it was found that the type of mosquito trap had a significant effect on collecting *Ae. aegypti* females (*p* < 0.001). Figure 2 illustrates that a higher number of females was collected using the BG-Sentinel trap when compared to those collected using the MosHouse trap (69.25 ± 17.19 vs. 30.75 ± 17.19).

### 3.2. Aedes aegypti Male Sampling with MosHouse Trap or MosHouse Trap with Sugar Stick—Comparing Irradiated vs. Non-Irradiated

In this experiment, we found that only the type of experimental mosquito, i.e., non-irradiated vs. irradiated, showed a significant effect, whereas the type of mosquito trap, i.e., MosHouse trap vs. MosHouse trap with sugar attractant, showed no significant effect on collecting *Ae. aegypti* males. The results from the two-way ANOVA showed that the type of experimental mosquito was a significant parameter (*p* < 0.001) (Table 2) affecting the collection of *Ae. aegypti* males, while the type of mosquito trap was not (*p* = 0.094). However, when focused on the interaction of the type of experimental mosquito and the type of mosquito trap in collecting *Ae. aegypti* males, these two factors showed no significant effect (*p* = 0.959). When focused on the original MosHouse trap, the results from the two-way ANOVA also showed that the type of experimental mosquito significantly affected *Ae. aegypti* male collection (*p* = 0.004). Significantly higher numbers of irradiated *Ae. aegypti* males were collected using the original MosHouse trap when compared to those of non-irradiated males (60.50 ± 5.45 vs. 50.50 ± 19.02) (Figure 3). The same results were also obtained when using the MosHouse trap with a sugar stick as an attractant. The type of experimental mosquito showed significant effect on collecting *Ae. aegypti* males (*p* = 0.005). Significantly higher numbers of irradiated *Ae. aegypti* males were collected using the MosHouse trap with a sugar stick when compared to those of non-irradiated males (64.50 ± 5.26 vs. 54.75 ± 10.05) (Figure 3).

### 3.3. Aedes aegypti Male Sampling with MosHouse Trap vs. MosHouse Traps with Sugar Stick—Effect of Sugar Stick on Capture of Irradiated or Non-Irradiated Males

In this study, we found that only the type of experimental mosquito (non-irradiated vs. irradiated) showed a significant effect on collecting *Ae. aegypti* males, whereas the type of mosquito trap (MosHouse trap vs. MosHouse trap with sugar attractant) showed no significant effect. The results from the two-way ANOVA showed that the type of experimental mosquito (*p* < 0.001), but not the type of mosquito trap (*p* = 0.170), significantly affected the collection numbers (Table 3). However, when focused on the interaction of the type of experimental mosquito and the type of mosquito trap in collecting *Ae. aegypti* males, these two factors showed no significant effect (*p* = 0.799). When focused on non-irradiated *Ae. aegypti* males, it was found that the type of mosquito trap showed no significant effect on collecting non-irradiated males (*p* = 0.436). Quite similar numbers of non-irradiated *Ae. aegypti* males were collected using a MosHouse with a sugar stick when compared to those collected using a MosHouse alone (54.75 ± 10.05 vs. 52.00 ± 19.49) (Figure 4). The same results were observed in irradiated *Ae. aegypti* males when the type of mosquito trap showed no significant effect on collecting irradiated males (*p* = 0.243). Quite similar numbers of irradiated *Ae. aegypti* males were collected using a MosHouse with a sugar stick when compared to those collected without a sugar stick (64.50 ± 5.26 vs. 60.50 ± 5.45) (Figure 4). 

### 3.4. Improved Aedes aegypti Male Sampling with MosHouse Trap—Effect of Internal Sugar Stick and External Sticky Flag

In this study, we found that the type of mosquito trap (MosHouse trap vs. MosHouse with sugar attractant vs. MosHouse with sticky flag) had a significant effect on collecting *Ae. aegypti* males. The results from the two-way ANOVA showed that the type of mosquito trap significantly affected the collection of *Ae. aegypti* males in the semi-field condition (*p* < 0.001) (Table 4). However, when focused on the MosHouse trap and the MosHouse trap with sugar attractant, the results showed that the type of mosquito trap had no significant effect on collecting males (*p* = 0.436). The mean numbers of *Ae. aegypti* males collected by using the MosHouse with sugar stick were not significantly different when compared to those collected using the original MosHouse trap (54.75 ± 10.05 vs. 52.00 ± 19.49) (Figure 5). However, when focused on the MosHouse trap and the MosHouse trap with sticky flags, it was found that the type of mosquito trap significantly affected the collection of *Ae. aegypti* males (*p* < 0.001). Higher numbers of *Ae. aegypti* males were collected when the double-sided sticky flags were added to the MosHouse when compared to those collected using the original MosHouse trap (75.25 ± 18.06 vs. 52.00 ± 19.49) (Figure 5).

### 3.5. Aedes aegypti Male and Female Sampling Using MosHouse Trap with External Sticky Flags—Effect of Sexes Released Separately

In this experiment, we found that the type of mosquito trap and the interaction of the type of mosquito trap and the sex of *Ae. aegypti* mosquitoes significantly affected the collection of *Ae. aegypti* males and females when they were released separately in a semi-field condition. The results from the two-way ANOVA showed that the type of mosquito trap (*p* < 0.001), but not the sex of mosquitoes (*p* = 0.068), significantly affected the collection of *Ae. aegypti* (Table 5). However, the interaction between these two factors significantly affected the collection of *Ae. aegypti* males and females (*p* < 0.001) (Table 5). When focused on *Ae. aegypti* males, it was found that the type of mosquito trap significantly affected the collection of males (*p* < 0.001). Higher numbers of *Ae. aegypti* males were collected using the MosHouse trap with additional sticky flags when compared to those of the MosHouse trap with only a stick panel (68.00 ± 16.12 vs. 7.25 ± 4.50) (Table 5). On the contrary, the type of mosquito trap significantly affected the collection of *Ae. aegypti* females (*p* < 0.001). A higher number of *Ae. aegypti* females was collected using the original MosHouse trap when compared to those collected using the MosHouse trap with additional sticky flags (41.75 ± 21.99 vs. 25.75 ± 10.24) (Figure 6).

### 3.6. Aedes aegypti Male and Female Sampling Using MosHouse Trap with External Sticky Flags—Effect of Sexes Released Simultaneously

In this experiment, we found that both the type of mosquito trap and the sex of *Ae. aegypti* mosquitoes, and the interaction of these two factors significantly affected the collection of *Ae. aegypti* males and females when they were released simultaneously. The results from the two-way ANOVA showed that the type of mosquito trap (*p* < 0.001) (Table 6), the sex of *Ae. aegypti* (*p* < 0.001), and their interaction significantly affected the collection of *Ae. aegypti* males and females (*p* < 0.001). When focused on *Ae. aegypti* males, it was found that the type of mosquito trap significantly affected the collection of males (*p* < 0.001). Higher numbers of *Ae. aegypti* males were collected using the MosHouse trap with additional sticky flags when compared to those collected using the MosHouse trap with only a sticky panel (46.75 ± 3.59 vs. 1.25 ± 1.26) (Figure 7). The same results were also observed when focused on *Ae. aegypti* females, i.e., the type of mosquito trap significantly affected the collection of females (*p* < 0.001). Higher numbers of *Ae. aegypti* females were collected using the MosHouse trap with additional sticky flags when compared to those collected using the MosHouse trap with only a sticky panel (28.00 ± 6.16 vs. 11.50 ± 4.65) (Figure 7).

## 4. Discussion

To highlight our findings, we demonstrated that there was no significant difference in the number of *Aedes aegypti* males, but a significantly lower number of females were collected using the original MosHouse trap when compared to those collected using the reference BG-Sentinel trap. The additional sticky flags significantly increased the efficiency of the original MosHouse trap, followed by the sugar sticks as an attractant. The application of MosHouse traps could be adjusted by an addition of sticky flags with or without sugar sticks in order to increase the collection of *Ae. aegypti* male mosquitoes in the field. When males or females were released separately, the MosHouse with additional sticky flags increased the efficiency of the original MosHouse trap for male collection as significantly higher numbers of males were collected. This was not the case for releasing females without males, since fewer females were collected using the MosHouse with additional sticky flags when compared to those collected using the original one with only a sticky panel. However, when males and females were simultaneously released, the MosHouse with additional sticky flags was more efficient in collecting both males and females of *Ae. aegypti* mosquitoes.

In this study, we first directly compared the efficiency of the BG-Sentinel trap and the MosHouse trap in collecting adult *Ae. aegypti* at the proportion of a one-to-one trap ratio in a semi-field condition. During the experiments, it was observed that a higher number of *Ae. aegypti* males and females collected in the BG-Sentinel trap was due to the sucking power of the electric fan motor when the mosquitoes flew close to the trap. Since the BG-Sentinel trap, equipped with the electric fan motor, had to be continuously connected to the batteries or a power source in order to be fully functional [22,23], it was not practical to leave the trap running for longer than 24 h without the intervention of an operator. In our experiment, we observed that when the batteries were low or out of power, the collected mosquitoes inside the collection bag flew out of the trap.

There is strong evidence demonstrating that *Ae. aegypti* mosquitoes are frequent sugar-feeders [41]. Sucrose sources and sugar-based solution have already been used as an attractant for trapping mosquitoes [7,42], since adult mosquitoes of both sexes require sugar intake as an important nutritional source for survival, especially male mosquitoes whose survival and reproductive success requires access to sugar sources [43,44,45]. In order to maximize the mosquito collection, the sugar stick was added to the MosHouse trap as a lure to attract *Ae. aegypti* mosquitoes. Our results showed that when the sugar stick was added to the MosHouse trap, slightly higher numbers of *Ae. aegypti* males were collected when compared to those collected by the MosHouse without a sugar stick, although the difference was not significant. Sissoko et al. (2019) and Roslan et al. (2017) found that mosquitoes were highly attracted to sugar sources when they were in the field [7,41]. In addition, as males were smaller than females, they took smaller sugar meals and needed to seek nectar more often than females [41,46,47]. Therefore, the addition of a sugar attractant to the MosHouse trap could be beneficial for mosquito collection, especially male mosquitoes, in semi-field or field conditions.

Various trapping methods have been generally applied to collect female mosquitoes [22,23,28,35,37,38]. The MosHouse trap was originally developed to collect *Ae. aegypti* female mosquitoes using double-sided sticky panels—the females being attracted to the moisture and darkness inside the trap [48,49]. In our experiments, when the females were separately released in a semi-field condition where the MosHouse traps were placed (Table 5), they tended to enter the trap in an attempt to seek suitable places to rest and oviposit their eggs, and in doing so were stuck on the sticky panel; hence more females were collected using the MosHouse trap with only the sticky panel. Previous studies showed that female mosquitoes preferred to seek shelter and dark surfaces, and they were more attracted to a shaded and wind-protected environment than open ones [50,51,52]. Other studies also showed that female mosquitoes were attracted to the transparent sticky film or board, and together with the presence of a vapor attractant behind that film, it increased the number of female mosquitoes as it was a strong attractant for oviposition-site-seeking mosquitoes [53,54,55].

In another experiment, the sticky flags were placed at the corners on top of the MosHouse trap in order to attract more male mosquitoes. Our results demonstrated that *Ae. aegypti* males were significantly collected at higher numbers using the MosHouse trap with additional sticky flags when compared to those collected using the MosHouse trap with a sticky panel alone. The reason that the highest numbers of male mosquitoes were collected using the MosHouse trap with additional sticky flags could be explained through the observation that when males mosquitoes flew around or passed over the MosHouse trap, they were easily stuck on the sticky flags placed outside of the trap, while it was more difficult for them to fly inside the trap and get stuck on the sticky panel where the sugar attractant was placed. Moreover, the design of the MosHouse trap was built with only one front entrance, making it even more difficult for the male mosquitoes to enter the trap. Johnson et al. (2017) showed that although the mosquito traps could attract males, there was no way to force them into the trap when they passed over; therefore, it was difficult to catch males in such a trap that was not equipped with tools to force them inside [56]. However, when *Ae. aegypti* females were released together with the males, the results showed that both females and males were significantly collected using the MosHouse trap with additional sticky flags. These results highlight the effect of additional sticky flags in increasing the efficacy of the original MosHouse trap with only a sticky panel, since significantly more *Ae. aegypti* males and females were collected when they were released together. This could be explained by the fact that when males were released together with females, they would fly around in order to search for females to mate [57,58,59,60] and then became stuck on the external sticky flags of the MosHouse traps. Our experiments demonstrated that both males and females were detected in significant numbers on the sticky flags that were placed outside the MosHouse trap, rather than on the sticky panel located inside the trap; and some were copulating pairs. Since these modified MosHouse traps can catch high numbers of *Aedes* male mosquitoes, they could be further refined as a male-specific trapping system [19,61].

The most studied versions of sticky traps are the double sticky trap and the MosquiTRAP (sticky ovitrap) [23,37,38]. The MosHouse trap is one example of a sticky trap with low cost, which could be deployed in large quantities to enhance the effectiveness of mosquito collection in the field [48,49]. Our study aimed to evaluate the efficiency of the sticky MosHouse trap that was built locally with inexpensive materials and was primarily used for collecting *Ae. aegypti* females for both surveillance and monitoring of dengue vector control programs. In addition, further modifications were made to the MosHouse, i.e., the addition of a sugar stick or sticky flags, in order to capture more males. The results from these semi-field experiments highlighted an increased efficiency of the MosHouse trap with additional sticky flags for collecting more *Ae. aegypti* males. However, a limitation of this study was the low number of experimental observations, as we could only perform an indirect comparison of the efficiency of the MosHouse trap vs. the MosHouse trap with some additional modifications. Therefore, further experiments of the MosHouse trap should be conducted in the field.

The absence of an efficient and sensitive collection method for the large-scale sampling of adult mosquito vectors is a major drawback to the epidemiological surveillance program, as well as the evaluation of the impact of control strategies and the surveillance of the spread of mosquito vectors into non-endemic regions [28,62]. Moreover, sampling methods for mosquito vectors that can provide more reliable entomological indicators of arbovirus disease transmission are essential [63]. Therefore, the development of an ideal operational trap that can be used to collect adult mosquito vectors in order to monitor their densities for vector surveillance as well as an evaluation of vector control methods has thus been considered a valuable contribution toward the prevention and control of mosquito-borne diseases [28,62]. As the MosHouse trap is low-cost, light-weight, easy to use, and requires neither power sources nor large spaces for its installation and transportation, it could be considered as an alternative trapping method to be used in combination with other mosquito-collection tools for vector surveillance and monitoring of vector control strategies in the field, as demonstrated in the pilot *Ae. aegypti* surveillance and control programs [48,49], or in large-scale applications, especially area-wide mosquito control programs using released sterile males.

## 5. Conclusions

When directly comparing the efficiency of the MosHouse traps with the BG-Sentinel traps, we found no significant difference in the numbers of collected males, while significantly lower numbers of females were observed with the MosHouse trap. We also found that sterilizing males by radiation significantly increased their collection when using MosHouse traps. Improvements to the MosHouse trap by adding a sugar stick and sticky flags could increase male collection—the latter increasing the number of males but not females when they were released separately, while the number of both males and females increased when they were released together. Therefore, the MosHouse trap was proved to be efficient and could be used as an alternative collecting tool in *Ae. aegypti* surveillance and control programs.

## Figures and Tables

**Figure 1 insects-13-01050-f001:**
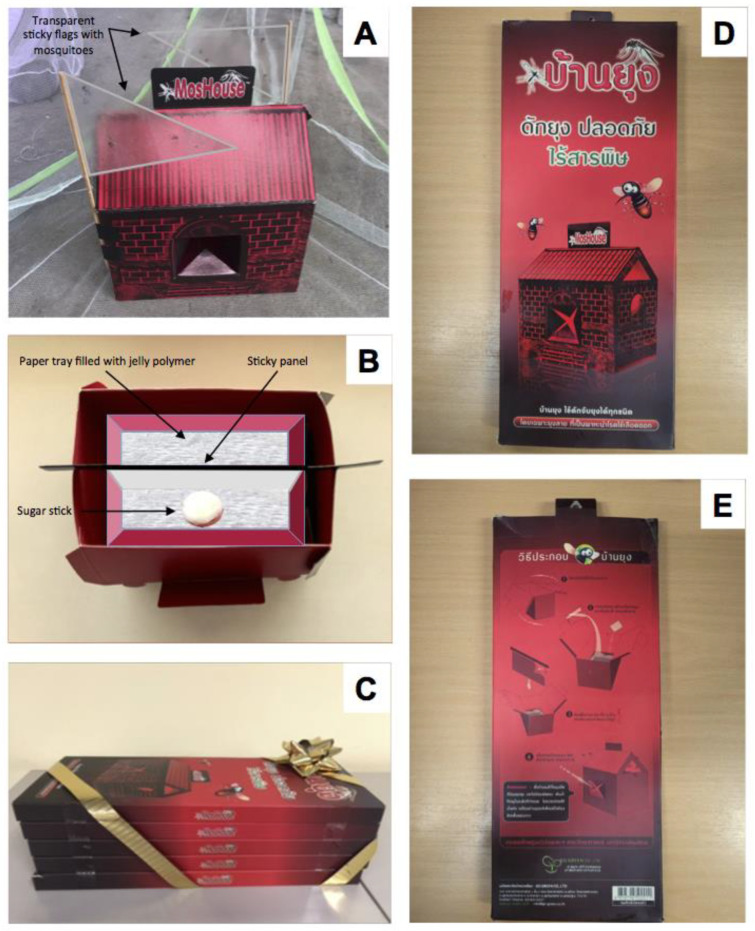
Pictures showing (**A**) a MosHouse trap with additional transparent double-sided sticky flags, (**B**) a MosHouse trap with a hanging double-sided sticky panel, a paper tray filled with jelly polymer and an additional sugar stick, (**C**) a package of five MosHouse traps, each with dimensions of 13.5 cm wide × 32.5 cm long × 2.0 cm high and weight of 153.37 g, (**D**) the front of a MosHouse package (English translation: MosHouse: Trap Mosquitoes; Safe; No Toxic Substances), and (**E**) the back of a MosHouse package (English translation: Assembly Method for MosHouse).

**Figure 2 insects-13-01050-f002:**
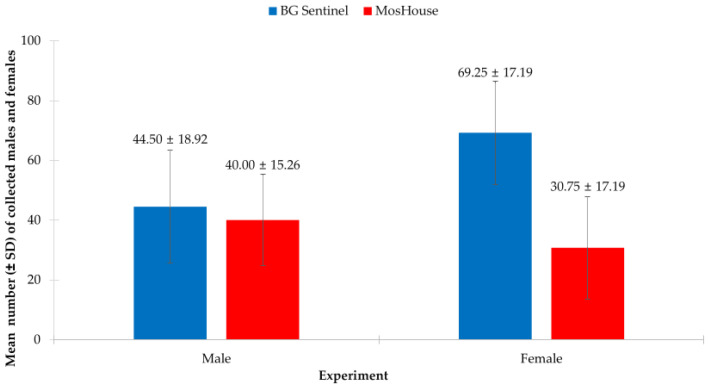
Comparison of the numbers of *Aedes aegypti* males and females (mean ± SD) collected using the BG-Sentinel traps vs. the MosHouse traps in a semi-field condition.

**Figure 3 insects-13-01050-f003:**
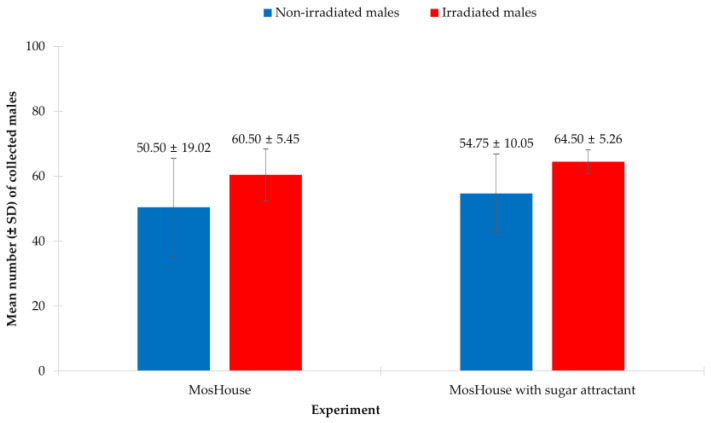
Comparison of the mean numbers of non-irradiated vs. irradiated *Aedes aegypti* males (mean ± SD) collected using the original MosHouse trap or the MosHouse trap with a sugar stick as an attractant in a semi-field condition.

**Figure 4 insects-13-01050-f004:**
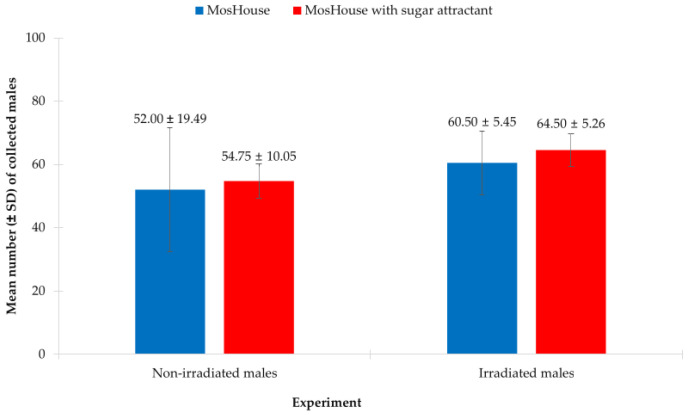
Comparison of the mean numbers of non-irradiated or irradiated *Aedes aegypti* males (mean ± SD) collected using the original MosHouse traps or the MosHouse traps with a sugar stick as an attractant in a semi-field condition.

**Figure 5 insects-13-01050-f005:**
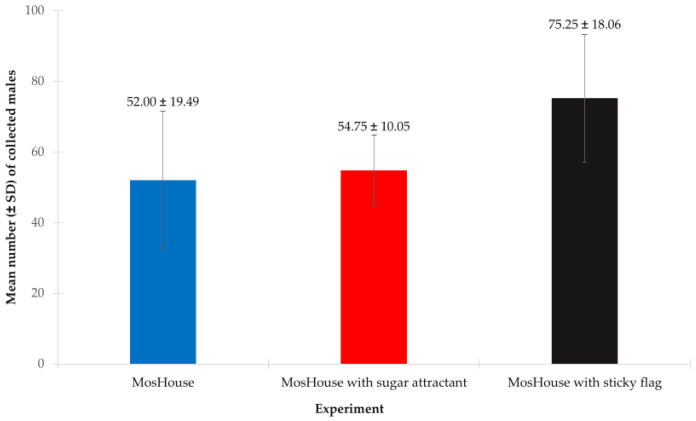
Comparison of the mean numbers of *Aedes aegypti* males (mean ± SD) that were collected using the original MosHouse traps, the MosHouse traps with a sugar stick as an attractant, or the MosHouse traps with additional sticky flags in a semi-field condition.

**Figure 6 insects-13-01050-f006:**
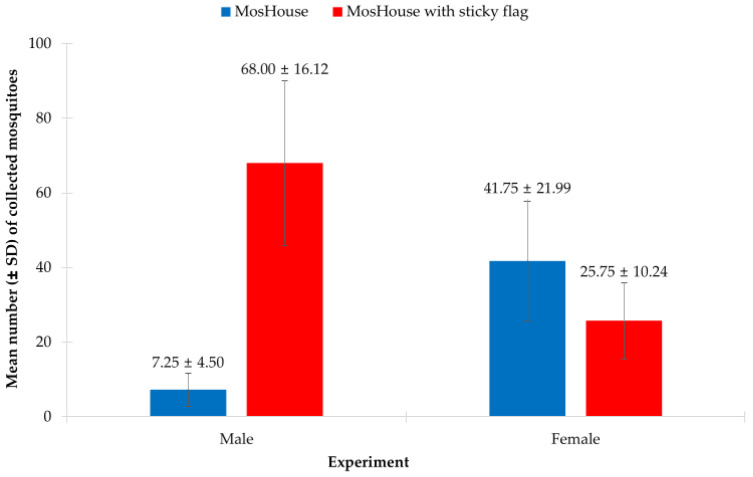
Comparison of the mean percentage of *Aedes aegypti* males and females (mean ± SD) collected using the original MosHouse trap and the MosHouse trap with additional sticky flags, when they were released separately in a semi-field condition.

**Figure 7 insects-13-01050-f007:**
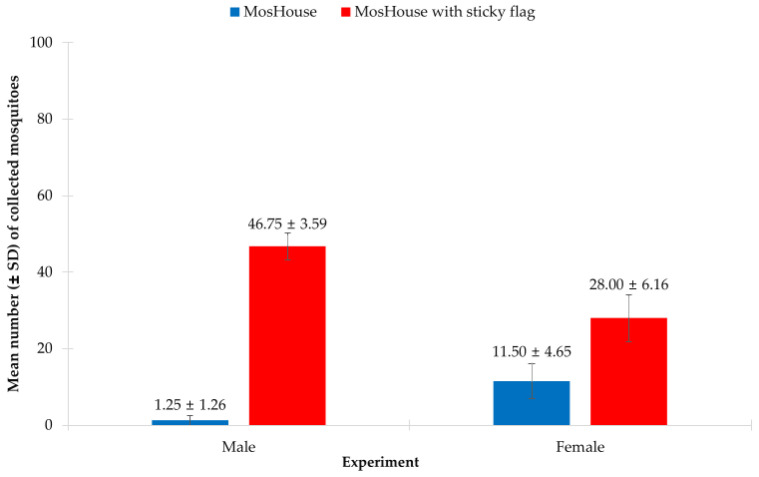
Comparison of the mean percentage of *Aedes aegypti* males and females (mean ± SD) collected using the original MosHouse trap and the MosHouse trap with additional sticky flags, when they were simultaneously released in a semi-field condition.

**Table 1 insects-13-01050-t001:** Results of two-way ANOVA on the numbers of *Aedes aegypti* males and females that were collected out of 100 mosquitoes released in the screened outdoor tents in Chachoengsao Province, eastern Thailand using the BG-Sentinel traps or the MosHouse traps. Interaction of the mosquito trap, the sex of *Ae. aegypti* mosquitoes, and the position of the mosquito trap (*p* < 0.001) were the main significant effects found in this experiment.

Source	Type III Sum of Squares	Df	Mean Square	F	*p*-Value
Corrected model	123.095 ^a^	15	8.206	38.877	<0.001
Intercept	680.805	1	680.805	3225.239	<0.001
Trap	36.980	1	36.980	175.189	<0.001
Sex	4.805	1	4.805	22.763	<0.001
Position	14.750	3	4.917	23.292	<0.001
Trap vs. Sex	23.120	1	23.120	109.528	<0.001
Trap vs. Position	21.325	3	7.108	33.675	<0.001
Sex vs. Position	14.010	3	4.670	22.124	<0.001
Trap vs. Sex vs. Position	8.105	3	2.702	12.799	<0.001
Error	672.100	3184	0.211		
Total	1476.000	3200			
Corrected total	795.195	3199			

^a^ refers to R-Square = 0.155 (Adjusted R-square = 0.151); dependent variable = collection number of *Ae. aegypti*; ‘Trap’ denotes the type of mosquito trap (BG-Sentinel trap vs. MosHouse trap); ‘Sex’ denotes the sex of *Ae. aegypti* mosquitoes (male vs. female); ‘Position’ denotes the position of the mosquito trap (1 vs. 2 vs. 3 vs. 4).

**Table 2 insects-13-01050-t002:** Results of the two-way ANOVA on the numbers of irradiated vs. non-irradiated *Aedes aegypti* males that were collected out of 100 mosquitoes released in the screened outdoor tents in Chachoengsao Province, eastern Thailand using the MosHouse traps or the MosHouse traps with a sugar stick. There was no significant interaction of the type of mosquito trap and the type of experimental *Ae. aegypti* male (*p* = 0.959) in this experiment.

Source	Type III Sum of Squares	Df	Mean Square	F	*p*-Value
Corrected model	4.582 ^a^	3	1.527	6.311	<0.001
Intercept	530.151	1	530.151	2190.504	<0.001
Trap	0.681	1	0.681	2.812	0.094
Exp	3.901	1	3.901	16.117	<0.001
Trap vs. Exp	0.001	1	0.001	0.003	0.959
Error	386.267	1596	0.242		
Total	921.000	1600			
Corrected total	390.849	1599			

^a^ refers to R-Square = 0.012 (Adjusted R-square = 0.010); dependent variable = collection numbers of *Ae. aegypti* males; ‘Trap’ denotes the type of mosquito trap (MosHouse trap vs. MosHouse with sugar attractant); ‘Exp’ denotes the type of experimental *Ae. aegypti* male (non-irradiated vs. irradiated).

**Table 3 insects-13-01050-t003:** Results of the two-way ANOVA on the collection numbers of irradiated or non-irradiated *Aedes aegypti* males that were collected out of 100 mosquitoes released in the screened outdoor tents in Chachoengsao Province, eastern Thailand using the MosHouse traps vs. the MosHouse traps with a sugar stick. There was no significant interaction of the type of mosquito trap and the type of experimental *Ae. aegypti* male (*p* = 0.799).

Source	Type III Sum of Squares	Df	Mean Square	F	*p*-Value
Corrected model	3.802 ^a^	3	1.267	5.238	0.001
Intercept	537.081	1	537.081	2220.000	<0.001
Trap	0.456	1	0.456	1.883	0.170
Exp	3.331	1	3.331	13.767	<0.001
Trap vs. Exp	0.016	1	0.016	0.065	0.799
Error	386.118	1596	0.242		
Total	927.000	1600			
Corrected total	389.919	1599			

^a^ refers to R-Square = 0.010 (Adjusted R-square = 0.008); dependent variable = collection numbers of *Ae. aegypti* males; ‘Trap’ denotes the type of mosquito trap (MosHouse trap vs. MosHouse with sugar attractant); ‘Exp’ denotes the type of experimental *Ae. aegypti* male (non-irradiated vs. irradiated).

**Table 4 insects-13-01050-t004:** Results of the two-way ANOVA on the collection numbers of *Aedes aegypti* males that were collected out of 100 mosquitoes released in the screened outdoor tents in Chachoengsao Province, eastern Thailand using the MosHouse traps vs. the MosHouse traps with a sugar stick vs. MosHouse traps with sticky flags. The mosquito traps were the main significant effect (*p* < 0.001) found in this experiment.

Source	Type III Sum of Squares	Df	Mean Square	F	*p*-Value
Corrected model	12.912 ^a^	2	6.456	28.261	<0.001
Intercept	441.653	1	441.653	1933.399	<0.001
Trap	12.912	2	6.456	28.261	<0.001
Error	273.435	1197	0.228		
Total	728.000	1200			
Corrected total	286.347	1199			

^a^ refers to R-Square = 0.045 (Adjusted R-square = 0.043); dependent variable = collection numbers of *Ae. aegypti* males; ‘Trap’ denotes the type of mosquito trap (MosHouse trap vs. MosHouse trap with sugar attractant vs. MosHouse trap with sticky flags).

**Table 5 insects-13-01050-t005:** Results of the two-way ANOVA on the collection numbers of *Aedes aegypti* males and females that were collected out of 100 mosquitoes released in the screened outdoor tents in Chachoengsao Province, eastern Thailand using the MosHouse traps or the MosHouse traps with sticky flags when they were released separately. The interaction of the mosquito traps and the sex of mosquitoes (*p* < 0.001) was the main significant effect found in this experiment.

Source	Type III Sum of Squares	Df	Mean Square	F	*p*-Value
Corrected model	79.532 ^a^	3	26.511	147.070	<0.001
Intercept	203.776	1	203.776	1130.464	<0.001
Trap	20.026	1	20.026	111.094	<0.001
Sex	0.601	1	0.601	3.332	0.068
Trap vs. Sex	58.906	1	58.906	326.784	<0.001
Error	287.693	1596	0.180		
Total	571.000	1600			
Corrected total	367.224	1599			

^a^ refers to R-Square = 0.217 (Adjusted R-square = 0.215); dependent variable = collection numbers of *Ae. aegypti*; ‘Trap’ denotes the type of mosquito trap (MosHouse trap vs. MosHouse trap with sticky flags); ‘Sex’ denotes the sex of *Ae. aegypti* (male vs. female).

**Table 6 insects-13-01050-t006:** Results of the two-way ANOVA on the collection numbers of *Aedes aegypti* males and females that were collected out of 100 mosquitoes released in the screened outdoor tents in Chachoengsao Province, eastern Thailand using the MosHouse traps or the MosHouse traps with sticky flags when they were released simultaneously. The significant interaction of the type of mosquito trap and the sex of mosquitoes (*p* < 0.001) was the main effect found in this experiment.

Source	Type III Sum of Squares	Df	Mean Square	F	*p*-Value
Corrected model	95.145 ^a^	3	31.715	248.158	<0.001
Intercept	153.125	1	153.125	1198.147	<0.001
Trap	76.880	1	76.880	601.558	<0.001
Sex	1.445	1	1.445	11.307	0.001
Trap vs. Sex	16.820	1	16.820	131.610	<0.001
Error	101.730	796	0.128		
Total	350.000	800			
Corrected total	196.875	799			

^a^ refers to R-Square = 0.483 (Adjusted R-square = 0.481); dependent variable = collection numbers of *Ae*. *aegypti*; ‘Trap’ denotes the type of mosquito trap (MosHouse trap vs. MosHouse trap with sticky flags); ‘Sex’ denotes the sex of *Ae*. *aegypti* (male vs. female).

## Data Availability

All data were provided in this manuscript and in the Appendix A.

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
