# Peer review of "The MosHouse® Trap: Evaluation of the Efficiency in Trapping Sterile Aedes aegypti Males in Semi-Field Conditions"

_insects, 2022, doi:10.3390/insects13111050_

Round 1
Reviewer 1 Report
This manuscript describes a valuable advancement in trapping mosquitoes with particular note to use of irradiated males. The experimental design is excellent, analysis detailed, conclusions sound and the manuscript well written. There are only a few minor editorial concerns that have been noted below.
Line 165, granules
Line 173 prolonging
Line 208, line 233, 239, 253, 268 species should be in italics
In your figures you have bars around the means but it does not indicate anywhere if these are SEM or SD. Please indicate which either on the figure or in the caption.
‘Figure 2’ and ‘Figure 3’ are in italics but other figure captions are not. Also you are not consistent for Tables
Figure 2, 3, Table 1, and 2 should have species name in italics. Also in Table 1,5 and 2 footnotes.
In the figure, the labels for the different colors (on the top of the figure) have a small font that is gray and not black. It would be much easier to read if the font was larger and the text was black. The same is true for the axis labels, it would be nicer if they were larger and darker.
Line 574-5
‘have’ should be ‘has’. As this is a summary, you indicate that radiation has a significant effect, but could you please say how it affects them (i.e., increase or decrease).
Reference 2 and 9 have page numbers in bold?
Reference 50 add a space after the year
References 14,27,32,51,52,58 and 59. Use complete numbers for page range similar to the other references
Author Response
Response to Reviewer 1 Comments:
The authors would like to thank the Reviewer for the useful and positive comments that help improving our manuscript. We have made corrections according to the comments as follows:
Line 165: “granule” was changed to “granules”
Line 173: “prolong’” was changed to “prolonging”
Lines 208, 233, 239, 253, 268 and others: Aedes aegypti was italicized.
Figures 2, 3, 4, 5, 6: SD was added in the Bar graph and the captions indicated “(mean ± SD)”.
Figures 2, 3 and Table 2: The word “Figure” and “Table” were changed to non-italic.
Figures 2, 3, Tables 1, 2 and Tables 1, 5 footnotes: Aedes aegypti was italicized.
Figures 2, 3, 4, 5, 6, 7: The front was enlarged and was changed to black.
Lines 574-5: “have” was changed to “has”.
References 2 and 9: The page numbers were changed to un-bold.
Reference 50: A space was added after the year.
References 14, 27, 32, 51, 52, 58, 59: The page numbers were completed.
Reviewer 2 Report
The authors in this manuscript report the enhanced attractively of low-cost traps for mosquitoes using multiple semi-field experiments. I think it is interesting for experts of the field, though it needs some refinement.
Statistical analysis:
1_ I am not a fan of Anova analysis for this type of data, I think that the authors might be better analyse all the data building a model (which would include all the explanatory variables). However, if the authors want to stick with this analysis, it would need more attention: a_they should state if the data are normally or quasi-normally distributed. b_ a p value of 0 it does not make sense. I think SPSS tells it only to say that the probability is really low, below 0.001; therefore I suggest to use p<0.001 instead of p=0.000.
2_ Be careful if the Aedes aegypti is italicized in the MS (for example line 233, 239, 253, 268, 288, 312, etc..).
3_ Paragraph 3.3: I do not understand if the data presented here are the same data from 3.2 and analysed in a different way...it is confusing.
Author Response
Response to Reviewer 2 Comments:
The authors would like to thank the Reviewer for the very useful comments, especially those that related to statistic analysis.
Point 1: We think that our data have not yet enough for building the model and we do not have the expertise to do it so we decided to stick with the Two-Way ANOVA. We have made corrections according to your comments:
a) Statistical analysis: The sentence was modified, “The number of mosquitoes collected from each experiment was quasi normally distributed and it was analysed by using the Two-way ANOVA.”
b) p = 0.000 was changed to p < 0.001 through out the manuscript.
Point 2: The word Aedes aegypti was italicized through out the manuscript.
Point 3: Some of the data in 3.3 was the same as 3.2 but they were analysed focusing different aims. 3.2 were analysed to compare the difference in the numbers of non-irradiated vs irradiated mosquitoes collected using MosHouse traps. 3.3 were analysed to compare the difference in the numbers of mosquitoes collected using original MosHouse vs MosHouse with additional sugar stick. The graphs and the text were modified to reflect the difference between the two experiments.
Reviewer 3 Report
Fascinating research. My only suggestion would be to include a comment in the discussion section regarding the use of Mostrap for vector surveillance. It seems to me that the usefulness of developed Mostrap for epidemiological and entomological aims is very close
Author Response
Response to Reviewer 3 Comments:
The authors would like to thank the Reviewer for the positive comments and the suggestions relate to the addition in the Discussion on the topic of vector surveillance using the MosHouse trap.
We have moved repetitive sentences related to vector surveillance from the Introduction to the Discussion and have made some additions as follows:
“The absence of an efficient and sensitive collection method for large-scale sampling of adult mosquito vectors is a major drawback to the epidemiological surveillance program, as well as the evaluation of the impact of control strategies and the surveillance of the spread of mosquito vectors into non-endemic regions [28,64]. Moreover, sampling methods for mosquito vectors that can provide more reliable entomological indicators of arbovirus disease transmission are essential [65]. Therefore, the development of an ideal operational trap that can be used to collect adult mosquito vectors in order to monitor their densities for vector surveillance as well as an evaluation of vector control methods has thus been considered a valuable contribution toward the prevention and control of mosquito-borne diseases [28,64]. As the MosHouse trap is low-cost, light-weight, easy to use, and requires neither power sources nor big space for its installation and transportation, it could be considered as an alternative trapping method to be used in combination with other mosquito collection tools for vector surveillance and monitoring of vector control strategies in the field as demonstrated in the pilot Ae. aegypti surveillance and control programs [49,50], or in the large scale applications especially the area-wide mosquito control programs using released sterile males.”
Reviewer 4 Report
The manuscript describes a series of experiments testing the effectiveness of the MosHouse trap, with and without various additional elements, and the BG sentinel trap in trapping male and or female Aedes aegypti mosquitoes.
The text is repetitive especially in the introduction and results sections and should be condensed. The grammar should also be improved. I have indicated some of the errors, but there are many more. I suggest the that the authors seek appropriate help.
Many of the references cited in the text are not in the correct order - for example reference 19 cited on line 119 refers to reference 20. This makes it almost impossible to evaluate the paper.
The trap needs to be described in more detail - plant infusion - gel etc.
In numerous places species names are not italicised.
Figure 1 - the photographs could be improved. The sticky flag and panel do not appear to be part of the photo. They look as though they have been added graphically. Are the flag and panel white or transparent surfaces? The paper tray and gel are not visible.
The numbers of mosquitoes trapped in the different experiments/replications should be included in a supplementary table. Also the numbers of individuals captured on the internal and external sticky surfaces should be provided. Also from the results it appears that the external sticky flag alone would have been much more effective in trapping males than the house trap. It would have been interesting to include additional experiments to test this.
Line 14: So far, mosquito....
Line 19: delete 'one'
Line 21: collected using...
Line 24: number of trapped males....
Line 34: collected using....
Line 37: male collection was achieved with....
Line 43: showing the potential
Lines 58-75: These two paragraphs are quite repetitive and should be condensed.
Lines 77-78: reference
Line 85: reference 8 does not appear to be the correct reference as it only considers sticky traps.
Line 95: Reference 21 does not consider sentinel traps with or without CO2.
Line 96: Reference 23 is not the correct reference.
Line 106: Reference 27 is not the correct reference.
Line 108: ease
Line 119: reference 19 should be reference 20
Line 153: container using ...
Line 154: into different containers prior to release ...
Line 162: plant infusion ? This should be expanded upon.
Line 164: What is the role of the jelly polymer? Humidity?
Line 170: weight
Line 188: describe the sugar stick here not in the next paragraph
Line 210 and elsewhere: Unfed for how long? Were they provided water but no sugar? How old were they?
Line 248: Also those on the external flag I presume?
Line 481: 'proportion of one to one' - this is not clear - It refers to the type of trap, not to the sex of the mosquitoes.
Line 515: attracted to...
Line 538-545: This is unnecessarily long-winded. Just mention that the increased number of males and females captured on the flags was due, perhaps, to copulating pairs alighting on the sticky surface. If this was the reason, one would expect many of the individuals on the sticky surface to be in copula - was this so?
Author Response
Response to Reviewer 4 Comments:
The authors would like to thank the Reviewer for the very useful and detail comments and suggestions that help improving our manuscript.
In general, we have rewritten some parts of the manuscript and rechecked the reference numbers following the reviewer’s comments.
The word “plant infusion” was changed to “hay infusion” which is generally used as attractant in mosquito traps.
In addition, the species name, Aedes aegypti, was rechecked to be all italicized in through out the manuscript.
Figure 1: The photo quality was improved. Figure 1A, the sticky flags were outlined only at the border to show the transparency and attracted mosquitoes. Figure 1B, we have improved it to make the paper tray, the jelly polymer, the sugar stick and the stick panel inside the tray visible in order to show how they were positioned inside. It is difficult to demonstrate clearly using only the photos rather than to combine the photos with the graphic ones to demonstrate the location of each component inside the MosHouse trap. Therefore, we have combined the photo and the graphic in Figure 1B.
Supplementary tables: We have added the supplementary tables to provide the numbers of mosquitoes collected in each experiment.
The sticky flag alone may not be able to attract Ae. aegypti mosquitoes as they usually attract to the combination of color, humidity, smell and etc. that make it into the MosHouse trap rather than the external stick flags. However, the external sticky flags could help capture more mosquitoes, especially the males that fly around outside the trap as they do not need the dark place to oviposit their eggs like the females that tend to enter the trap for resting and egg oviposition.
Line 14: It was corrected. So far, mosquito…
Line 19 “one” was deleted.
Lines 21, 34: It was corrected through out the manuscript. …collected using…
Line 24: It was corrected. …number of trapped males…
Line 37: It was corrected. …male collected was achieved with…
Line 43: It was corrected. …showing the potential…
Line 58-75: The two paragraphs were condensed as suggested. Some sentences were moved to the Discussion section.
Introduction: “Monitoring the abundance of adult mosquito vectors by evaluating their distribution and density are important aspects for the development of control strategies [7,8]. Since the mosquito traps and trapping techniques that are commercially available have been deployed mostly to reduce nuisance populations of adult mosquitoes [5,9], efficient trapping of mosquito vectors is still urgently needed [2,10].”
Lines 77, 78: The references were added. “In general, Ae. aegypti mosquitoes, which are the main vector of dengue virus, breed and live inside households or within the surrounding environment [11,12]. In addition, they usually do not disperse very far from where they emerge given that blood meal sources are always available [13].”
Lines 85, 95, 96, 106, 119: The references were rechecked and corrections were made through out the manuscript.
Line 108. It was corrected. …the ease in directly identifying..
Line 153: It was corrected. …container using…
Line 154: It was corrected. …into different containers prior to release…
Line 162: “plant infusion” was changed to “hay infusion”.
Line 164: The purpose of the jelly polymer was added in the text as follows: …jelly polymer (to provide humidity after absorbing water)…
Line 170: It was corrected …very light weight…
Line 188: The description of sugar stick was added earlier as suggested. “3) whether the sugar stick which is the cotton soaked with 10% sugar solution could increase the capture rate of irradiated and non-irradiated Ae. aegypti males;…”
Line 210 and elsewhere: The addition of the age of mosquitoes and the unfed period were added through out the manuscript where it was relevant. “100 of 3-5 days old, 24-hour unfed, non-irradiated Ae. aegypti males were released in the screened tents…”
Line 248: It was added as follows: “After 24 hours, the sticky panel hanging inside the MosHouse trap and also the external sticky flags were collected, and the number of males that entered into the MosHouse trap and got struck on the sticky panel and also those that got struck on the external sticky flags was determined.”
Line 481: It was modified where it was relevant through out the manuscripts as follows: …with a trap number of one to one ratio….
Line 515: It was corrected. …attracted to…
Line 538-545: It was modified as follows: “It could be explained by the fact that when males were released together with females, they would fly around in order to search for females to mate [58,59,60,61] and then got struck on the external sticky flags of the MosHouse traps. Our experiments demonstrated that both males and females were detected in significant numbers on the sticky flags that were placed outside the MosHouse trap, rather than on the sticky panel located inside the trap; and some were copulating pairs.” In fact, we have seen some copulating pairs on the sticky flags. One example may be unclearly seen on Figure 1A.
Thank you so much for your time and effort to help us with this manuscript.
Round 2
Reviewer 4 Report
The authors have improved the manuscript. However, I suggest that they correct the following:
Title: “Conditions”
Line 38: “Significantly higher”
Line 42: “efficiently as the”
Line 134: “less maintenance, etc., for male collection.”
Line 162: “Males and females”
Line 189: “were tested in 5 m x 5 m screened tents”
Line 196: “which consists of cotton soaked with …”
Line 205: “Secondly, …”
Line 221: “Mosquitoes were then captured using…”
Line 237: “24-hour unfed, …”
Line 238: ”same conditions”
Line 241: “collected using portable”
Lines 242-244: delete
Line 245: “traps”
Line 247: “non-irradiated males”
Line 259: “.. traps was placed …”
Line 263: “Any male mosquitoes that remained in the screened tents were collected using portable vacuum aspirators.”
Lines 265-267: delete
Line 269: “effect of sexes released separately”
Line 275: “Any male mosquitoes that remained in the screened tents were collected using portable vacuum aspirators.”
Lines 280-282: delete
Line 284: “effect of sexes released simultaneously”
Line 291 “Any mosquitoes that remained in the screened tents were collected using portable vacuum aspirators.”
Lines 294-296: delete
Line 309: “Figure 2 illustrates the lack of significant differences in the numbers…”
Line 313: “Figure 2 illustrates that a higher number of females collected using the BG….collected using the MosHouse..”
Line 336, 346, 357, 366, 387 & 390: “experimental”
Line 347: “Significantly higher…”
Line 348: “non-irradiated males ..”
Line 370: delete “was”
Line 380: “numbers”, “were collected”
Line 381: “those collected”
Line 427: “sexes”
Line 438: “On the contrary,”
Line 489: “a significantly lower number of females”
Line 498: “since fewer females”
Line 510: “without the intervention of an operator.”
Line 576: “mosquitoes, they could be further refined as a male-specific trapping system”
Line 580: “quantities”
Line 611: “significantly”
Author Response
Responses to Reviewer 4 Comments:
The authors would like to thank the reviewer for his/her kindness to correct the gammas/mistakes of the manuscript that makes it improved significantly.
Title: “Condition” was changed to “Conditions”.
Line 38: “significant higher” was changed to “significantly higher”.
Line 42: “efficient as the” was changed to “efficiently as the”.
Line 134: “less maintenance, and etc., for male collection.” Was changed to “less maintenance, etc., for male collection.”.
162: “Male and females” was changed to “Males and females”.
189: “was tested in a 5 m x 5 m screened tents” was changed to “were tested in 5 m x 5 m screened tents”.
Line 196: “which is the cotton soaked with…” was changed to “which consists of cotton soaked with…”.
Line 205: “Second…” was changed to “Secondly…”
Line 221: “Mosquitoes were then collected back by using…” was changed to “Mosquitoes were then captured using…”.
Line 237: “24-hour,…” was changed to “”24-hour unfed,…”
Line 238: “same condition” was changed to “same conditions”
Line 241: “collected by using portable…” was changed to “collected using portable…”.
Lines 242-244: were deleted as suggested.
Line 245: “trap” was changed to “traps”.
Line 247: “non-irradiated male” was changed to “non-irradiated males”.
Line 259: “…trap was placed” was changed to “traps was placed…”.
Line 263: “Any male mosquitoes that remaining in the screened tents were collected by using portable vacuum aspirators.” Was changed to “Any male mosquitoes that remained in the screened tents were collected using portable vacuum aspirators.”.
Lines 265-267: were deleted as suggested.
Line 269: “effect of gender released separately” was changed to “effect of sexes released separately”.
Line 275: Any male mosquitoes that remaining in the screened tents were collected by using portable vacuum aspirators.” was changed to “Any male mosquitoes that remained in the screened tents were collected using portable vacuum aspirators.”.
Lines 280-282: were deleted as suggested.
Line 284: “effect of gender released simultaneously” was changed to “effect of sexes released simultaneously”.
Line 291: “Any mosquitoes that remaining in the screened tents were collected by using portable vacuum aspirators.” was changed to “Any mosquitoes that remained in the screened tents were collected using portable vacuum aspirators.”.
Lines 294-296: was deleted as suggested.
Line 309: “Figure 2 showed no significant differences in the numbers…” was changed to “Figure 2 illustrates the lack of significant differences in the numbers…”
Line 313: “Figure 2 showed higher number of females collected by using BG…collected by using the MosHouse…” was changed to “Figure 2 illustrates that a higher number of females collected using the BG…collected using the MosHouse…”
Lines 336, 346, 357, 366, 387 & 390: “experimented” was changed to “experimental”
Line 347: “Significant higher…” was changed to “Significantly higher…”
Line 348: “non-irradiated male…” was changed to “non-irradiated males…”
Line 370: “was” was deleted as suggested.
Line 380: “number”, “was collected” was changed to “numbers, “were collected”
Line 381: “those without a sugar stick…” was changed to “those collected without a sugar stick…”
Line 427: “gender” was changed to “sexes”
Line 438: “In contrary” was changed to “On the contrary”
Line 489: “a significant lower number of females” was changed to “a significantly lower number of females”
Line 498: “since less females” was changed to “since fewer females”
Line 510: “without caring by an operator” was changed to “without the intervention of an operator”
Line 576: “mosquitoes, it could be potential benefits to be further developed into male-specific trapping system” was changed to “mosquitoes, they could be further refined as a male-specific trapping system.”
Line 580: “quantity” was changed to “quantities”
Line 611: “significant” was changed to “significantly”